# Altered Ocular Surface Health Status and Tear Film Immune Profile Due to Prolonged Daily Mask Wear in Health Care Workers

**DOI:** 10.3390/biomedicines10051160

**Published:** 2022-05-18

**Authors:** Sharon D’Souza, Tanuja Vaidya, Archana Padmanabhan Nair, Rohit Shetty, Nimisha Rajiv Kumar, Anadi Bisht, Trailokyanath Panigrahi, Tejal S. J., Pooja Khamar, Mor M. Dickman, Ruchika Agrawal, Sanjay Mahajan, Sneha Sengupta, Rudy M. M. A. Nuijts, Swaminathan Sethu, Arkasubhra Ghosh

**Affiliations:** 1Department of Cornea and Refractive Surgery, Narayana Nethralaya, Bangalore 560010, India; drrohitshetty@narayananethralaya.com (R.S.); drtejal@narayananethralaya.com (T.S.J.); dr.poojakhamar@narayananethralaya.com (P.K.); drsnehasengutpa@gmail.com (S.S.); 2GROW Research Laboratory, Narayana Nethralaya Foundation, Bangalore 560099, India; tanuja.vaidya@narayananethralaya.com (T.V.); archana.nair@narayananethralaya.com (A.P.N.); nimisha@narayananethralaya.com (N.R.K.); anadi@narayananethralaya.com (A.B.); trailoknath@narayananethralaya.com (T.P.); ruchika.agarwal12830@gmail.com (R.A.); sanjay@narayananethralaya.com (S.M.); 3Manipal Academy of Higher Education, Udupi 576104, India; 4University Eye Clinic Maastricht, Maastricht University Medical Center, 6200 MD Maastricht, The Netherlands; mor.dickman@mumc.nl (M.M.D.); rudy.nuijts@mumc.nl (R.M.M.A.N.); 5MERLN Institute for Technology-Inspired Regenerative Medicine, Maastricht University, 6200 MD Maastricht, The Netherlands

**Keywords:** ocular surface discomfort, mask, COVID-19, tear fluid, soluble factors, nociception, immune cells, hypercapnia

## Abstract

Prolonged daily face mask wearing over several months might affect health of the ocular surface and is reported to be associated with complaints of discomfort and dry-eye-like symptoms. We studied the ocular surface clinical parameters, tear soluble factors and immune cell proportions in ophthalmologists practicing within similar environmental conditions (*n* = 17) at two time points: pre-face-mask period (Pre-FM; end of 2019) and post-face-mask-wearing period (Post-FM; during 2020 COVID-19 pandemic), with continuous (~8 h/day) mask wear. A significant increase in ocular surface disease index (OSDI) scores without changes in tear breakup time (TBUT), Schirmer’s test 1 (ST1) and objective scatter index (OSI) was observed Post-FM. Tear soluble factors (increased—IL-1β, IL-33, IFNβ, NGF, BDNF, LIF and TSLP; decreased—IL-12, IL-13, HGF and VEGF-A) and mucins (MUC5AC) were significantly altered Post-FM. Ex vivo, human donor and corneoscleral explant cultures under elevated CO_2_ stress revealed that the molecular profile, particularly mucin expression, was similar to the Post-FM tear molecular profile, suggesting hypercapnia is a potential contributor to ocular surface discomfort. Among the immune cell subsets determined from ocular surface wash samples, significantly higher proportions of leukocytes and natural killer T cells were observed in Post-FM compared to Pre-FM. Therefore, it is important to note that the clinical parameters, tear film quality, tear molecular factors and immune cells profile observed in prolonged mask-wear-associated ocular surface discomfort were distinct from dry eye disease or other common ocular surface conditions. These observations are important for differential diagnosis as well as selection of appropriate ocular surface treatment in such subjects.

## 1. Introduction

The COVID-19 pandemic caused by severe acute respiratory syndrome coronavirus 2 (SARS-CoV-2) was initially reported in December 2019 in Wuhan, China. It has proved to be one of the greatest global challenges in recent years not only due to the morbidity and mortality associated with the disease but also due to its far-reaching consequences and impact on other aspects of healthcare and lifestyle [1,2]. More than 500 million people have been infected around the world and the death toll has crossed 5.4 million as reported by the Johns Hopkins University Coronavirus Resource Center [3]. As the infection spreads primarily through aerosolisation and via fomites, usage of protective face masks over the nose and mouth has been one of the cornerstones of disease prevention [4,5]. Therefore, the importance of wearing face masks in public has been an integral part of all national and international medical recommendations pertaining to the pandemic [6,7]. Hospital staff and medical health professionals are at particular risk due to their continuous interaction with patients and their prolonged potential exposure [8]. Mandatory usage of face masks of different kinds and other personal protective equipment is one of the best ways to reduce the risk of acquiring and spreading the disease and hence is part of daily clinical practice across the world [9,10]. Different kinds of masks including N95 masks, respirators, surgical masks and cloth masks are used among health care professionals and the community [11]. However, an undesirable consequence of this prolonged mask usage has been the increase in subjects complaining of dry-eye-like symptoms [12]. This observation was termed mask-associated dry eye (MADE) in a previous report [13]. Such “dry-eye-like” symptoms are more likely to affect those who wear masks for prolonged periods of time out of necessity such as healthcare professionals, elderly patients and those with high-risk co-morbidities or individuals with predisposition to ocular surface issues such as those using digital screens extensively in air-conditioned settings [8]. This is an important public health consideration because the irritation and discomfort caused at the ocular surface due to prolonged mask wearing could induce frequent removal of the mask and eye rubbing, which could be counterproductive to health care requirements. However, it is yet to be determined whether this reported ocular surface condition truly fits the established profile of dry eye disease (DED).

A proposed causal hypothesis of such symptoms is that the exhaled air escaping upward from the edge of the mask simulates evaporative conditions on the ocular surface, drying the ocular surface and leading to inflammation and discomfort [13]. Another possibility is that as the exhaled air has higher concentrations of carbon dioxide [14], the hypercapnic air escaping upwards could contribute to the complaints and ocular surface changes as well. Hypercapnia is known to induce inflammation [15], but it has not been tested on the ocular surface. Although ocular surface symptoms [12,13,16] post mask wear have been reported, the associated alterations to tear film parameters or molecular profile of the ocular surface have not been investigated. As the current trends in the COVID-19 pandemic indicate the need for continued mask usage in the foreseeable future, we are likely to observe an increasing number of people with this problem. Thus, greater awareness of the presentation and the possible underlying mechanisms would help us treat this condition more appropriately. In this study, we evaluated the effect of prolonged mask wearing on the ocular surface clinical parameters, tear fluid soluble factors and ocular surface immune cell proportions in a tightly controlled group of subjects. We hypothesised that exhaled, CO_2_-rich air causes hypercapnia at the ocular surface, causing irritation, and tested the same in ex vivo human ocular surface (corneoscleral) explant culture and in human corneal epithelial cells in vitro.

## 2. Materials and Methods

### 2.1. Study Cohort

The study was approved by the Narayana Nethralaya institutional ethics committee (EC Ref. No: C/2021/04/01; 6 May 2021) and was conducted as per the tenets of the Declaration of Helsinki. All samples were collected from practicing ophthalmologists who volunteered with written informed consent. The study design included investigating ocular surface clinical parameters, tear soluble factors and ocular surface immune cell proportions of the same study volunteers (*n* = 17; age 30 ± 4.2 years; M/F–9/8) at two different time points. The first time point for clinical parameters, sample collection, processing and documentation was performed at the end of 2019 (pre-COVID-19 pandemic) termed as the pre-face-mask-wearing period (Pre-FM). The subjects included were part of routine clinical practice in the outpatient department in an ophthalmology tertiary care centre which did not require continuous (approximately 8 h a day) wearing of face masks during the first time point of the study. The second time point for data and sample collection from the same study subjects was at the end of 2020 (COVID-19 pandemic period), termed as the post-face-mask-wearing period (Post-FM), where subjects were required to have continuous (approximately 8 h a day for over 6 months) wearing of face masks as per guidelines during the COVID-19 pandemic period. All subjects in the study used standard N95 masks. Only subjects with paired clinical evaluation and samples (pre and post 6-month mask wear) were included in this study. Clinical parameters included ocular surface disease index (OSDI) score, Schirmer’s test 1 (ST1), tear breakup time (TBUT), tear film interferometry using the Lipiview (TearScience Inc. Morrisville, NC, USA ) and objective scatter index (OSI) on the Optical Quality Analysis System II (OQAS, Visiometrics, Terrassa, Spain) which were measured along with collection of tear fluid and ocular surface wash from the same subjects during Pre-FM and Post-FM between 10 a.m. and 12 noon. The ocular surface disease index (OSDI) questionnaire score is a recognised standard for evaluating patient symptoms of dry eye [17]. The ST1 was performed for 5 min without topical anaesthesia, using a sterile Schirmer’s test strip. TBUT was measured three times consecutively after fluorescein instillation and the median value was recorded. Corneal and conjunctival staining was evaluated after instillation of fluorescein (Contacare Ophthalmics and Diagnostics, Vadodara, Gujarat, India). OQAS is a double-pass aberrometer and allows objective measurement of ocular image quality [18]. Subjects with pre-existing symptoms, ocular surface disease, dry eye, contact lens usage, usage of chronic ocular medications, usage of any topical medications in the preceding 6 weeks, history of previous ocular surgery and active ocular infection were excluded. It is to be noted that none of the study subjects tested positive for COVID-19, nor did they report any symptoms associated with COVID-19 during the study period.

### 2.2. Tear Fluid Collection

Tear fluid samples were collected from the study subjects using Schirmer’s strips (Contacare Ophthalmics and Diagnostics, Vadodara, Gujarat, India) by following Schirmer’s test I protocol. Briefly, the blunt end of the sterile Schirmer’s strip was folded at the notch by 90° to form a hook-shaped bend. The bent tip of the Schirmer’s strip was placed gently into the inferior-temporal aspect of the conjunctival sac (lower eyelid) of the subject’s eye using sterile forceps. The Schirmer’s strip was then allowed to be wetted by the subject’s tear fluid via capillary action. The wetting length in millimetres was measured by observing the wetting end against graduation/scale marks on the strip at the end of 5 min. The Schirmer’s strip was then collected and stored in a sterile microcentrifuge tube at −80 °C until further processing. Tear fluid samples were collected from the study subjects using Schirmer’s strips by following Schirmer’s test I protocol and were stored in microcentrifuge tubes at −80 °C until further processing. Tear proteins were extracted from Schirmer’s strips by agitation in 300 µL of sterile 1 × PBS for 2 h at 4 °C as previously described [19]. The tear fluid was eluted by centrifugation and was stored at −80 °C until further analyses.

### 2.3. Tear Soluble Factor Measurements

The levels of interleukin (IL) IL-1α, IL-1β, IL-2, IL-6, IL-8, IL-9, IL-10, IL-12/IL-23p40, IL-13, IL-17A, IL-21, IL-33, interferon (IFN) IFNα, IFNβ, IFNγ, tumour necrosis factor (TNF) alpha, CX3CL1/fractalkine, CXCL1/GROα, CXCL10/IP-10, CXCL11/I-TAC, CCL2/MCP1, CXCL9/MIG, CCL5/RANTES, BDNF, NGF, HGF, TGFβ1, VEGF, sICAM1, sVCAM, sL-selectin, sP-selectin, sIL-1R1, sIL-1R2, sIL-2Ra, sTNFRI, sTNFRII, LIF, angiogenin, NGAL, granzymes, perforins and TSLP in the tear fluid were measured simultaneously using multiplex ELISA as described earlier [19]. IL-1α, IL-2, IL-6, IL-8, IL-9, IL-10, IL-12/IL-23p40, IL-13, IL-17A, IL-21, IFNα, IFNγ, TNFα, CX3CL1/fractalkine, CXCL10/IP-10, CXCL11/I-TAC, CCL2/MCP1, CXCL9/MIG, CCL5/RANTES, TGFβ1, VEGF, sICAM1, sVCAM, sL-selectin, sP-selectin, sIL-1R1, sIL-1R2, sIL-2Ra, sTNFRI, sTNFRII and angiogenin were measured using Cytometric Bead Array (BDTM CBA Human Soluble Protein Flex Set System, BD Biosciences, USA) on a flow cytometer (BD FACSCantoII, BD Biosciences, San Jose, CA, USA). BD FACSDiva software (BD Biosciences, San Jose, CA, USA) was used to acquire the beads and record signal intensities. FCAP array version 3.0 (BD Biosciences, San Jose, CA, USA) was used to determine the absolute concentration of the analytes using respective standards. Similarly, IL-1β, IL-33, IFNβ, CXCL1/GROα, BDNF, NGF, HGF, LIF, NGAL, granzymes, perforins and TSLP were measured by multiplex ELISA using the LEGENDplex kit (BioLegend Inc., San Diego, CA, USA) according to the manufacturer’s instructions and were measured on a flow cytometer (BD FACS CantoII, BD Biosciences, San Jose, CA, USA). BD FACSDiva software (BD Biosciences, San Jose, CA, USA) was used to acquire the beads and record signal intensities. Absolute concentration was determined based on respective standards using LEGENDplex™ Data Analysis Software Suite (BioLegend Inc., San Diego, CA, USA). The absolute concentrations of these analytes were obtained using respective standards using GraphPad Prism 6.0 (GraphPad Software, Inc., La Jolla, CA, USA). The wetting length of the Schirmer’s strip during tear collection and tear elution buffer volume was used to calculate the dilution factor to derive the normalised concentration of the tear analytes.

### 2.4. Ocular Surface Immune Cell Collection

The ocular surface wash (OSW) collected by a trained clinician was used to isolate and identify the different immune cells on the ocular surface. The ocular surface was gently irrigated in the outpatient department. The saline was collected from the lateral canthus of the eye in a sterile tube and the cells were fixed by adding 0.05% paraformaldehyde. The samples were stored at 4 °C until processing.

### 2.5. Ocular Surface Immune Cell Phenotyping by Flow Cytometry

The different immune cell subsets on the ocular surface and their proportions were measured using immunophenotyping by flow cytometry as described previously [20]. Fluorochrome-conjugated antibodies specific to the various immune cell subtypes are as follows: CD45-APC-H7, CD11b-BV510, CD14-FITC, CD16-BV605, CD66b-AF647, CD56-PECy7, CD3-PE, CD19-PerCP and CD138-APC-R700. Immune cell subsets were identified using a manual gating strategy (Appendix A).

### 2.6. Cell Culture and Explant Culture

Human Corneal Epithelium HCE2 (ATCC^®^ CRL-11135™) was cultured in F-12 DMEM media (Gibco, Waltham, MA, USA) containing 10% fetal bovine serum (Gibco, Waltham, MA, USA) and 1% antibiotic and antimycotic mixture (Gibco, Waltham, MA, USA) at 37 °C. The normocapnia (5% CO_2_) and hypercapnia (20% CO_2_) conditions were maintained in Thermo Scientific™ Forma™ Steri-Cycle™ CO_2_ Incubators as described previously [15]. Human corneoscleral rims from healthy donors (*n* = 6) undergoing corneal transplantation were collected in MK medium after removal of the central cornea (which is used for keratoplasty). Corneal rims were washed for 30 s with 1% antibiotic/antimycotic (Gibco, Waltham, MA, USA), followed by 1xPBS twice. Each corneoscleral rim was cut into equal halves and maintained under normocapnia or hypercapnia for 24 h in F-12 DMEM media (Gibco, Waltham, MA, USA) supplemented with 10% FBS (fetal bovine serum, Gibco, Waltham, MA, USA) and 1% antibiotics (Gibco, USA) at 37 °C.

### 2.7. Osmolarity and pH Measurements

HCE2 cells were seeded on 24-well tissue culture plates (Corning, Glendale, AZ, USA) and exposed to either normocapnia (5% CO_2_) or hypercapnia (20% CO_2_) for 24 h. Culture wells without cells were kept as internal controls for each condition. In total, 100 µL of media was collected from each sample and the osmolarity was measured using a Knauer K-7400S Semi-micro Osmometer (Berlin, Germany) as per manufacturer’s recommendation. Ex vivo culture media from human corneoscleral rims kept at either normocapnia or hypercapnia conditions were measured for osmolarity using Knauer K-7400S Semi-micro Osmometer. In addition, pH of the media was also measured and no differences were observed (data not included) pre and post CO_2_ exposure.

### 2.8. Quantitative PCR

Post termination of incubation period, HCE2 cells and/or corneoscleral rims were harvested and washed with ice-cold PBS. Total RNA was extracted from the cells using 500 μL of TRIZOL reagent (Invitrogen, Carlsbad, CA, USA) according to the manufacturer’s instructions, followed by total RNA isolation and cDNA conversion using iScript cDNA synthesis (Bio-Rad, Philadelphia, PA, USA). The quantitative real-time PCR cycle included pre-incubation at 95 °C for 5 min, 40 amplification cycles at 95 °C for 10 s, 60 °C for 15 s and 72 °C for 30 s using a CFX Connect real-time PCR detection system (Bio-Rad, Philadelphia, PA, USA). Gene expression of target genes was analysed and reported after normalisation to actin.

### 2.9. Mucin Measurement

The levels of mucins (MUC5AC, MUC16) were measured in the tear fluid of study subjects and the supernatant of the corneoscleral rim explant culture using enzyme-linked immunosorbent assay—ELISA (MUC5AC, E-EL-H2279, Elabscience, Houston, TX, USA; MUC16, E-EL-H0636, Elabscience, Houston, TX, USA) according to the manufacturer’s protocol.

### 2.10. Statistical Analysis

All data were assessed for normality using the Shapiro–Wilk normality test. The differences between sample groups were analysed using Wilcoxon matched-pairs signed rank test or paired *t*-test (GraphPad Prism 8.0, Inc., La Jolla, CA, USA). Spearman rank correlation test was performed using MedCalc^®^ Version 12.5 (MedCalc Software, Ostend, Belgium) to determine the association between the test parameters. All group data are presented in bar graphs or tables as mean ± standard error of mean (SEM). *p* < 0.05 is considered to be statistically significant.

## 3. Results

The clinical parameters and tear molecular factors were assessed from data collected prior to COVID-19 (Pre-FM) and after 6 months of regular mask wear (Post-FM).

### 3.1. Ocular Surface Clinical Indices

A significant (*p* < 0.0001) increase in the OSDI scores was observed in subjects during the period that involved face mask wear for an extended period of time (Post-FM) compared with the respective OSDI score recorded during the pre-MASK-wearing period (Pre-FM) as shown in Figure 1a. A significant (*p* < 0.0001) increase in both discomfort scale and vision scale during Post-FM compared with Pre-FM contributed to the increased OSDI score in the study subjects (Figure 1). However, we observed an increase in Schirmer’s test 1 (ST1) and tear breakup time (TBUT) scores (Figure 1b,c). No significant differences in the other tear quality parameters such as objective scatter index (Figure 1d,e) and lipid content (Figure 1f) were observed in the study subjects between the two time periods. This suggests that the study subjects had an increase in symptoms but did not develop any detrimental change in clinical signs associated with ocular surface health. Instead, there was a consistent increase in tear quality parameters, which did not explain the high OSDI scores.

### 3.2. Tear Soluble Factors Indicate a Skewed Inflammatory Profile Post Mask Wear

Tear fluid soluble factors have often been studied to determine molecular contributors associated with various ocular surface conditions. Since usage of mask in our cohort presented with symptoms without the classical signs of ocular surface disease, we sought to understand if there was a molecular explanation for this observation. Therefore, the levels of a range of soluble factors that include cytokines, chemokines, soluble cell adhesion molecules, soluble receptors, enzymes, etc., were measured in Pre-FM and Post-FM matched tear samples. The levels of IL-1α, IL-1β, IL-2, IL-33, IFNβ, LIF, TSLP, BDNF, NGF, perforins, sTNFRI and sIL-2Rα were observed to significantly increase in the tear fluid of samples of Post-FM compared with Pre-FM (Figure 2; Appendix A). On the contrary, the tear fluid levels of IL-6, IL-8, IL-13, IL-18, I-TAC, RANTES, granyzmes, HGF, VEGF, sVCAM, sL-selectin, sIL-1R1, sIL-1R2 and sTNFRII were significantly lower in the Post-FM compared with Pre-FM samples (Figure 3; Appendix A). These results indicate that classical inflammation-associated molecular factors were not elevated or even reduced post mask wear in these subjects. However, a subset of inflammatory factors and pain- or nociception-related factors was elevated.

### 3.3. Ocular Surface Immune Cell Profile Reveals Significant Changes Due to Mask Wear

Altered ocular surface immune cell proportions and phenotypes have been reported during ocular surface homeostasis perturbations [20]. Hence, changes in ocular surface immune cell proportions were determined in ocular surface wash samples collected during pre-ME and post-ME. Significantly (*p* < 0.05) higher proportions of leukocytes (Figure 4a) and natural killer T cells (Figure 4g) were observed in Post-FM samples compared with Pre-FM (Figure 4). An increase (not statistically significant) in the proportion of T cells among the ocular immune cells (Figure 4f) compared with ocular surface neutrophils (Figure 4k) was also observed. A significant (*p* < 0.001) reduction in the proportions of eosinophils, B cells and antibody-producing B cells (plasma cells) was observed in the ocular surface wash during the mask-wearing era (Post-FM) compared with Pre-FM (Figure 4h–j). Clinical parameter status along with tear soluble factors and ocular surface immune proportion alterations suggest a unique imbalance in ocular surface health due to the extended use of face masks (Table 1).

### 3.4. Exposure to 20% CO_2_ Induces a Hypercapnic Response and Reduces Osmolarity in In Vitro and Ex Vivo Cultures

To test the theory that increased CO_2_ saturation might lead to a hypercapnic stress response, we subjected human corneal epithelial cell cultures to 20% CO_2_ (hypercapnia) for 24 h. In all the experiments, cells and media at 5% CO_2_ (normocapnia) served as controls. The cells exposed to hypercapnia did not show any morphological changes or induction of hyperosmolar stress following exposure to simulated hypercapnia (Appendix A). The osmolarity of the media was significantly reduced in both the cell-free and cell-containing condition when exposed to the hypercapnic condition (Figure 5a). We therefore used primary donor ex vivo corneoscleral rim explants for the next set of experiments which closely mimic the response from the human eye. Our measurements show a similar reduction in osmolarity in the media containing corneoscleral rims as well (Figure 5b). Therefore, we further analysed gene expression from these corneoscleral rims and found PHOX2B, a known indicator of ongoing hypercapnic stress, to be significantly elevated (Figure 5c) in the hypercapnia category.

### 3.5. Hypercapnia Altered Inflammatory Profile from Human Primary Ex Vivo Cultures

To determine the effect of hypercapnia on human eyes, we measured gene expression of a set of inflammation-related genes in corneoscleral rims. The expression of inflammatory factors such as IL-8 and TNFα was reduced post hypercapnia (Figure 5d–f) while IFNβ was elevated (Figure 5g), although the expression of IL-6 and VEGF-A was not altered (Figure 5d,h). It is to be noted that despite the trend that was observed, only reduction in expression of IL-8 was statistically significant. We further validated the levels of a select set of secreted factors in the corneoscleral rim explant culture media using multiplex ELISA assays. We observed IL-6, IL-8, TNFα, HGF and VEGF levels as being reduced but IL1α, IFNγ, LIF and perforins levels as being elevated in hypercapnic samples compared to normocapnic controls (Table 2). Though the values were not statistically significant, the trends were similar to those observed in the tear samples (Figure 2 and Figure 3).

### 3.6. Hypercapnia Is Associated with Increased Mucin Production

The tear film stability is dependent, in part, on the level of mucins produced [21,22]. The human cornea and conjunctiva express membrane-spanning mucins such as MUC16 and soluble mucins such as MUC5AC whose levels have been found to be deficient in DED [23]. Hypercapnia is known to increase mucin secretion [24,25]. We therefore assessed the tear mucin profile in a subset of the tears within our cohort. The change in MUC5AC and MUC16 levels varied widely between subjects. However, a mean increase in levels of both mucins was observed (Table 3a) post mask wear. Therefore, to clarify if hypercapnia may cause mucin secretion, we analysed, the levels of MUC5AC and MUC16 in corneoscleral rims by ELISA (in the supernatant) and gene expression. MUC5AC expression was elevated in all three donors, while MUC16 was elevated in two donors and the mean fold difference in hypercapnia was higher for both mucins (Table 3b; Figure 6). The summary of the results is represented in Figure 7.

## 4. Discussion

Mask-associated ocular-surface-related changes can have long-term implications for ocular health. There have also been recent reports of ocular surface issues in postsurgical patients which could be attributed to prior face mask usage [26]. This emphasises the need to recognise this condition early and treat it appropriately. While there are many theories about how mask usage affects the ocular surface, it is generally agreed that face masks oppose the outward movement of exhaled air through the mask, resulting in its partial upward expiration over the eyes. It is proposed that such airflow recreates an environment of faster tear evaporation and thereby causes “dry-eye-like” symptoms of irritation and discomfort [16]. This condition has also been known to occur in patients in intensive care units who are on different mechanical and positive pressure ventilation (CPAP) if there are air leaks over the ocular surface [27,28]. Taping the mask at the top has been suggested to reduce this upward air escape, but may also be detrimental as it can cause subtle changes in eye closure if not placed properly, thereby worsening existing symptoms [13].

DED has been classically associated with reduction in tear secretion as measured by Schirmer’s test and tear film stability as measured by the TBUT [29]. This is usually associated with different symptoms such as increased ocular discomfort and visual disturbances of varying severity [29]. In our study cohort of practicing ophthalmologists with a tightly controlled work environment, there was a significant increase in the OSDI score in all subjects, especially in the discomfort score. However, there was no reduction in either the ST1 values or the TBUT, which would have confirmed a diagnosis of DED. On the contrary, we found a small increase in ST1 and TBUT in the Post-FM measurements. One possible explanation for the increase in ST1 value could be tear hypersecretion and epiphora which can occur in patients with DED; however, in those cases, there is usually associated tear film instability and evaporative dry eye [30]. As TBUT was also slightly increased in our cohort, it rules out evaporative DED. Another method of assessing the tear film stability is by measuring the lipid layer thickness (LLT), which is reduced in evaporative dry eye [31]. In our cohort, there was no change in tear film interferometry and LLT. Optical quality can also be affected in patients with ocular surface disorders [32], and is easily measured using the OSI on the OQAS. There was no change in OSI among our cohort of subjects in the Post-FM measurements. Thus, we find an increase in the symptomatology of the subjects without corresponding changes in the objective tests for DED. This could imply a pathology that is dissimilar to DED. In addition, the small increase in Schirmer’s and TBUT cannot be explained by this diagnosis.

DED is known to be associated with a derangement in specific inflammation-related molecular factors [19,33]. These inflammatory markers have been found to be altered irrespective of the type of DED and are strongly correlated with the clinical features [34]. Increase in proinflammatory factors interleukin-1 (IL-1), IL-6, IL-8, IL-1 receptor antagonist (IL-1RA), MMP-9, chemokines and tumour necrosis factor-alpha (TNF-α), which attract and activate immune cells, has been reported in DED [34,35,36]. In our study, the classic DED-related inflammatory factors IL-6 and IL-8 were reduced in tears post MASK wear and a different group of molecular factors showed an increase. A number of proinflammatory factors including IL-1β, IL-33, TSLP, BDNF and NGF were observed to significantly increase in the Post-FM samples. However, the levels of anti-nociceptive factors such as IL-13, HGF and VEGF-A were significantly lower in the Post-FM group. In effect, there is an overall imbalance between pro- and anti-nociceptive factors in addition to an increase in inflammation on the surface, which may explain the increased discomfort in the subjects even in the absence of signs of DED. Previous studies have shown that an increase in pro-nociceptive factors and decrease in the levels of anti-nociceptive factors are associated with increased pain and discomfort in patients with DED [19,37]. Our data suggest a mechanism other than dry eye driving the increase in ocular surface discomfort and nociception post mask wear.

The ocular surface immune cell profile in DED has also been studied and was shown to have significantly higher proportions of leukocytes, neutrophils, CD4 T cells, CD8 T cells and CD4/CD8 T cells ratio [20]. Neutrophils/NK cells ratio was also significantly higher in DED [20,35]. In this study, there was a significant increase in the proportion of T cells compared to ocular surface neutrophils and a reduction in the proportions of eosinophils, B cells and antibody-producing B cells (plasma cells). Higher proportions of leukocytes and natural killer T cells Post-FM samples compared with Pre-FM were also noted. This alteration in the immune cell profile is distinct from that seen in DED and points to a different mechanism of action in this condition. As the molecular and immune cell alterations do not correlate with a dry eye scenario, an alternate mechanism was explored.

Exhaled air has a higher concentration of carbon dioxide (4–5%) compared to inhaled air (0.4%) [14]. There have also been studies which have evaluated the levels of carbon dioxide within the face mask and shown that there can be up to a 10-fold increase in the carbon dioxide levels within the reservoir formed between the face and the mask as well [38]. We hypothesised that the increased carbon dioxide in the exhaled air flowing over the ocular surface may induce hypercapnia-related changes and alter the molecular profile. Since we observed the TBUT slightly rising in Post-FM measurements, we tested the levels of mucins, a key component of tear film integrity, across samples.

Mucins such as MUC16 and MUC5AC are key components of the tear film and ocular surface [23]. Alterations in their levels can affect the tear film stability and have been found to be lower in tears of patients with DED [23]. These mucins are also secreted in other parts of the body including the lungs. In vivo experiments on pulmonary tissues have demonstrated that hypercapnia can increase mucin5AC secretion and induce inflammation [24,25]. A similar result was seen in the tears of our subjects which showed an increase in mucin levels post prolonged mask wear. In the ex vivo experiment on corneoscleral rim explants, hypercapnia induced the expression of MUC5AC and MUC16. The increase in TBUT and Schirmer’s values could therefore be attributable to an increase in mucin levels in the tears which enhanced tear film stability [21,22].

In addition, we found that the profile of inflammatory factor changes observed in the corneoscleral rims exposed to hypercapnia was similar to that observed in the tear samples from Post-FM group. This included a reduction in IL-6, IL-8, TNFα, HGF and VEGF-A, but an increase in IL-1α, IFNγ, LIF and perforins. The lack of statistical significance in these explant culture experiments could be due to the donor variations and the small number of corneoscleral rims studied. This could imply that hypercapnia is a key factor in the mask-induced ocular surface changes seen in these subjects. The PHOX2B gene expression served as a positive control for the cellular induction of hypercapnia [39]. Interestingly, we observed a reduction in osmolarity of the culture supernatants of cells and corneoscleral rims exposed to hypercapnia. This is contrary to the hyper-osmolarity associated with DED [21,40]. This feature further reiterates the possibility that the changes induced post prolonged face mask usage are distinct from DED. Taken together, our data present a scenario where prolonged mask wear creates a hypercapnic tear film, which has more mucin and increased stability, but has higher proportions of pro-nociceptive inflammatory factors leading to ocular surface discomfort. This could explain the increased OSDI scores in our subjects without corresponding clinical signs. The limitations of the study are the lack of a control group without mask wear, which was not possible in view of the health hazard, and multiple time points of measurement to delineate acute vs. chronic responses.

## 5. Conclusions

It is important to note that mask wear induces discomfort, with hypo-osmolarity on the ocular surface, suggesting that lubricant eye drop selection is important for symptomatic relief. Hypo-osmolar lubricating eye drops have been found to be effective in Sjogrens syndrome in a study comparing them to iso-osmolar lubricating eye drops [41]. However, these eye drops can be avoided in mask-associated ocular surface disease. Although mucin secretagogues have been shown to be effective in treating certain types of dry eye [23], it would be prudent to avoid them in these subjects in view of pre-existing increased mucin secretion. Therefore, iso-osmolar eye drops and anti-inflammatory or anti-nociceptive options may be considered. Preventive measures could include better fitting face masks which reduce the efflux of exhaled air over the eyes. Identification of this problem is very important and leading questions about the type and duration of mask usage should be part of the routine clinical history for a patient with symptoms of ocular discomfort during this pandemic.

The possibility of extended mask wear till the COVID-19 pandemic abates brings with it a possible surge in mask-wear-associated ocular surface issues. Hence, it is important for health care professionals to be aware of this condition and find ways to treat or prevent it. This study provides molecular evidence of hypercapnia-induced nociception as a potential mechanism for mask-associated ocular surface discomfort.

## Figures and Tables

**Figure 1 biomedicines-10-01160-f001:**
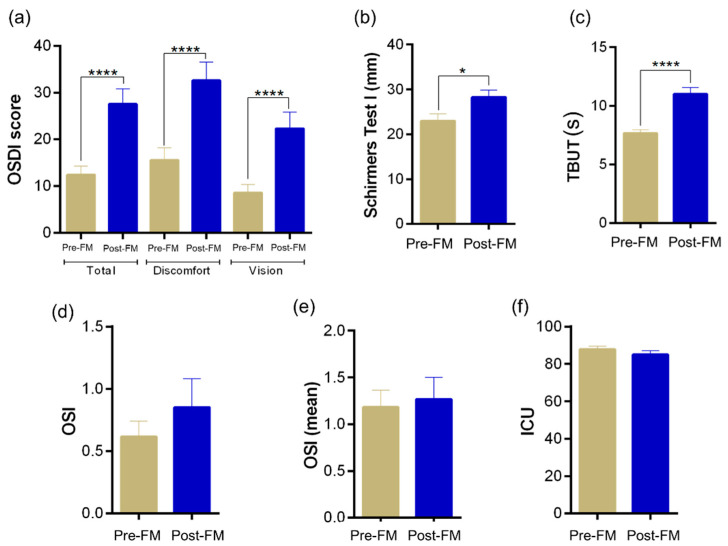
Status of ocular surface health indices in healthy individuals during the face-mask-wearing era. Bar graphs represent: (**a**) ocular surface disease index (OSDI) scores and (**b**–**f**) tear film quality assessments in matched study subjects prior to and during the face-mask-wearing era. (**b**) Schirmer’s test 1 values in mm/5 min, (**c**) tear breakup time—TBUT values in s (**d**) objective scatter index—OSI, (**e**) mean objective scatter index and (**f**) lipid layer thickness of the tear film quantified as interferometric colour units—ICU. *N* = 17; 34 eyes. Pre-FM indicates period where the study subjects were not wearing a face mask (pre COVID-19 era) for an extended period of time. Post-FM indicates period where the same study subjects were wearing a face mask (during the COVID-19 pandemic) for an extended period of time (approximately 8 hours a day). Bar graph indicates mean ± SEM. * *p* < 0.05, **** *p* < 0.0001, Wilcoxon matched-pairs signed rank test.

**Figure 2 biomedicines-10-01160-f002:**
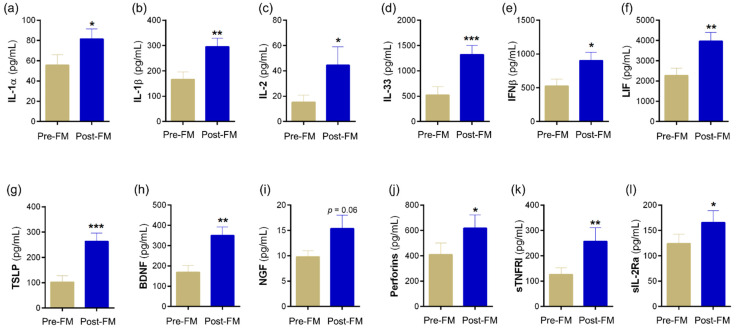
Tear fluid soluble factors increased during face-mask-wearing era. Bar graphs represent the levels of cytokines—(**a**) IL-1α, (**b**) IL-1β, (**c**) IL-2, (**d**) IL-33, (**e**) IFNβ, (**f**) LIF, (**g**) TSLP; growth factors—(**h**) BDNF, (**i**) NGF; cytolytic protein—(**j**) perforin; and soluble receptors—(**k**) sTNFRI, (**l**) sIL-2R α in matched study subjects prior to and during the face-mask-wearing era. IL—interleukin; IFNβ—interferon beta; LIF—leukaemia inhibitory factor, an IL-6 class cytokine; TSLP—thymic stromal lymphopoietin; BDNF—brain-derived neurotrophic factor; NGF—nerve growth factor; sTNFRI—soluble tumour necrosis factor receptor I; sIL-2Rα—soluble form of IL-2 receptor alpha. *N* = 17; 34 eyes. Pre-FM indicates period where the study subjects were not wearing a face mask (pre COVID-19 era) for an extended period of time. Post-FM indicates the period where the same study subjects were wearing a face mask (during the COVID-19 pandemic) for an extended period of time (approximately 8 hours a day). Bar graph indicates mean ± SEM * *p* < 0.05, ** *p* < 0.01, *** *p* < 0.001; Wilcoxon matched-pairs signed rank test.

**Figure 3 biomedicines-10-01160-f003:**
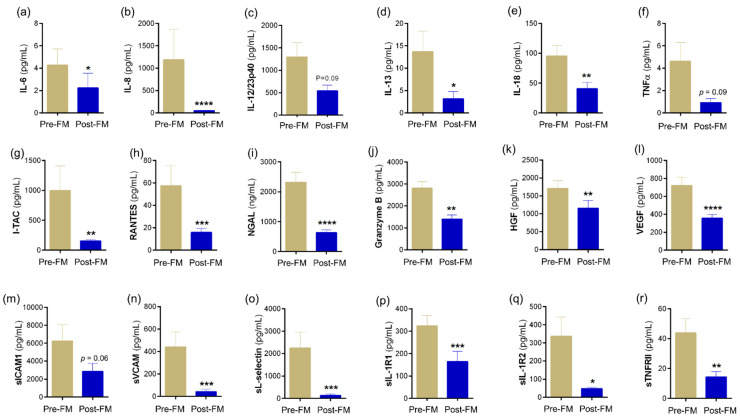
Tear fluid soluble factors decreased during face-mask-wearing era. Bar graphs represent the levels of cytokines—(**a**) IL-6, (**c**) IL-12/23p40, (**d**) IL-13, (**e**) IL-18, (**f**) TNFα; chemokine—(**b**) IL-8, (**g**) I-TAC, (**h**) RANTES; enzymes—(**i**) NGAL, (**j**) granzyme; growth factors—(**k**) HGF, (**l**) VEGF; soluble cell adhesion molecules—(**m**) sICAM1, (**n**) sVCAM, (**o**) sL-selectin; soluble receptors—(**p**) sIL-1R1, (**q**) sIL-1R2, (**r**) sTNFRII in matched study subjects prior to and during the face-mask-wearing era. IL—interleukin; TNFα—tumour necrosis factor alpha; I-TAC—interferon-inducible T-cell alpha chemoattractant, CXCL11; RANTES—regulated upon activation, normal T Cell expressed and presumably secreted, CCL5; NGAL—neutrophil gelatinase-associated lipocalin; HGF—hepatocyte growth factor; VEGF—vascular endothelial growth factor-A; sICAM1—soluble intercellular adhesion molecule-1; sVCAM—soluble vascular cell adhesion molecule; sIL-1R1—soluble IL-1 receptor type 1; sIL-1R1—soluble IL-1 receptor type 2; sTNFRII—soluble tumour necrosis factor receptor II; *N* = 17; 34 eyes. Pre-FM indicates period where the study subjects were not wearing a face mask (pre COVID-19 era) for an extended period of time. Post-FM indicates the period where the same study subjects were wearing a face mask (during the COVID-19 pandemic) for an extended period of time (approximately 8 hours a day). Bar graph indicates mean ± SEM * *p* < 0.05, ** *p* < 0.01, *** *p* < 0.001, **** *p* < 0.0001; Wilcoxon matched-pairs signed rank test.

**Figure 4 biomedicines-10-01160-f004:**
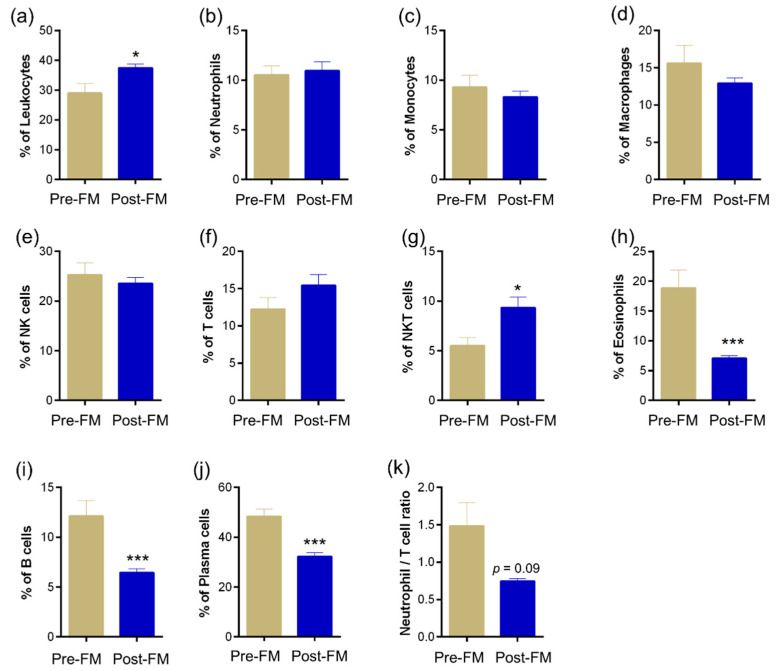
Ocular surface immune cell subset proportions during the face-mask-wearing era. Graph indicates percentage of (**a**) leukocytes—CD45+ cells within the cells analysed in ocular surface wash samples collected from matched study subjects prior to and during the face-mask-wearing era. Graphs indicate the percentages of (**b**) neutrophils—CD45+CD11b+CD16+CD66b+ cells, (**c**) monocytes—CD45+CD14+ cells, (**d**) macrophages—CD45+CD16+ cells, (**e**) natural killer cells—NK cells—CD45+CD16+CD56+ cells, (**f**) T cells—CD45+CD3+ cells, (**g**) NKT cells—CD45+CD3+CD16+CD56+ cells, (**h**) eosinophils—CD45+CD11b+CD16-CD66b+ cells, (**i**) B cells—CD45+CD3-CD19+ cells and (**j**) plasma cells—CD45+CD3-CD19+CD138+ cells within the leukocyte population in ocular surface wash samples collected from matched study subjects prior to and during the face-mask-wearing era. The neutrophil to lymphocyte ratio (**k**) was also calculated. *N* = 12; 24 eyes. Pre-FM indicates period where the study subjects were not wearing a face mask (pre COVID-19 era) for an extended period of time. Post-FM indicates period where the same study subjects were wearing a face mask (during the COVID-19 pandemic) for an extended period of time (approximately 8 hours a day). Bar graph indicates mean ± SEM * *p* < 0.05, *** *p* < 0.001; Wilcoxon matched-pairs signed rank test.

**Figure 5 biomedicines-10-01160-f005:**
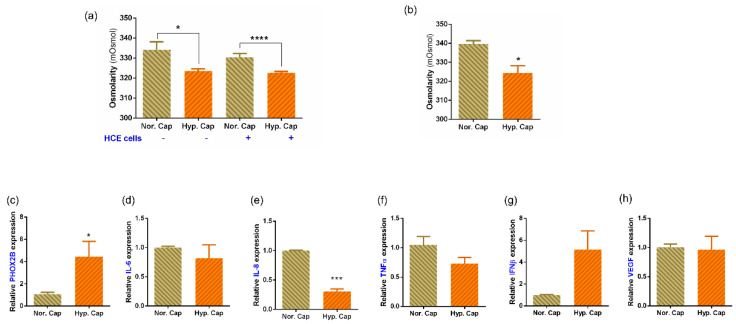
Effect of hypercapnia on osmolarity and expression of genes in human corneal epithelial monolayer culture and corneoscleral rim explant cultures. (**a**) Bar graph represents the osmolarity changes in culture media with and without SV40 immortalised human corneal epithelial cells (HCE2) following exposure to either 5% CO_2_ (normocapnia—Nor. Cap) or 20% CO_2_ (hypercapnia—Hyp. Cap), in vitro for a period of 24 h. Bar graph represents mean ± SEM of six biological replicates. * *p* < 0.05, **** *p* < 0.0001; two-tailed unpaired *t*-test. (**b**) Bar graph represents the osmolarity changes in culture media that contained corneoscleral rims following exposure to either 5% CO_2_ (normocapnia—Nor. Cap) or 20% CO_2_ (hypercapnia—Hyp. Cap), in vitro for a period of 24 h. Bar graph represents mean ± SEM from three matched biological replicates or corneoscleral rims. * *p* < 0.05, two-tailed paired sample *t*-test. (**c**–**h**) Graphs indicate relative mRNA expression of PHOX2B, IL-6, IL-8, TNFα, IFNβ and VEGF in matched corneoscleral rims following exposure to either 5% CO_2_ (normocapnia—Nor. Cap) or 20% CO_2_ (hypercapnia—Hyp. Cap), in vitro for a period of 24 h. The expression of PHOX2B, IL-6, IL-8, TNFα, IFNβ and VEGF was normalised to the expression of β-actin (housekeeping gene). Bar graph indicates mean ± SEM from two technical replicates for each of the three biological replicate experiments. * *p* < 0.05, *** *p* < 0.001; two-tailed paired sample *t*-test. PHOX2B—paired-like homeobox 2B.

**Figure 6 biomedicines-10-01160-f006:**
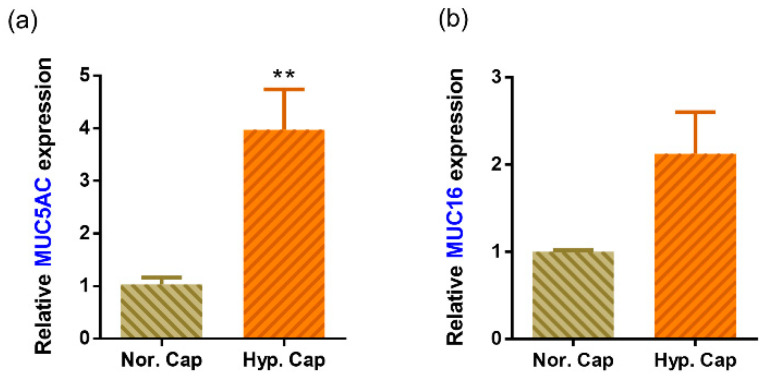
Effect of hypercapnia on the expression of mucins in human corneoscleral rim explant cultures. Graphs indicate mean relative mRNA expression of MUC5AC (**a**) and MUC16 (**b**) in matched corneoscleral rims following exposure to either 5% CO_2_ (normocapnia—Nor. Cap) or 20% CO_2_ (hypercapnia—Hyp. Cap), in vitro for a period of 24 h. The expression of MUC5AC and MUC16 was normalised to expression of β-actin (housekeeping gene). The bar graph indicates mean ± SEM from two technical replicates for each of the three biological replicate experiments. ** *p* < 0.01; two-tailed paired sample *t*-test.

**Figure 7 biomedicines-10-01160-f007:**
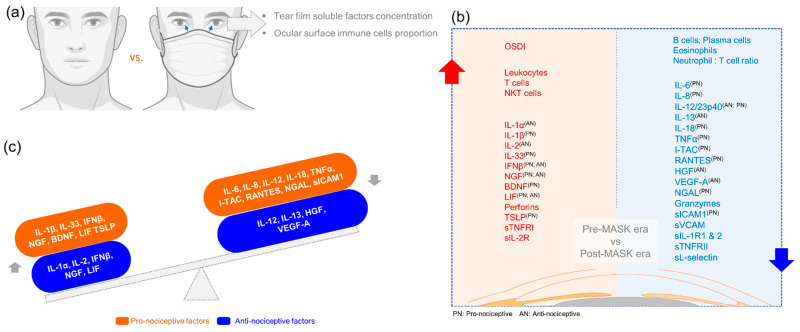
Schematic summary of the status of ocular surface clinical indices, proportion of immune cell subsets and tear fluid soluble factor levels during the face-mask-wearing era. Panel (**a**) is a schematic representation of the study design. Left side (red zone) of panel (**b**) summarises the list of clinical indices, immune cell subsets and tear soluble factors that increased during the post face mask era compared to the pre face mask era in matched subjects. Right side (blue zone) of panel (**b**) summarises the list of immune cell subsets and tear soluble factors that decreased during the post face mask era compared to the pre face mask era in matched subjects. Since ocular surface discomfort was one of the main clinical indices that was significantly high, we categorised the tear soluble factors based on their ability to modulate nociception in order to study the relationship between tear soluble factors and the discomfort experienced by the study subjects during the post face-mask-wearing era. Superscript of PN and/or AN beside the various analytes indicates whether they possess pro-nociceptive (PN) or anti-nociceptive (AN) potential as reported in the literature and not limited to ocular surface pain. Panel (**c**) demonstrates that both pro- and anti-nociceptive factors were dysregulated in tear fluid during post face-mask-wearing era. However, it can be posited that an increase in a select set of pro-nociceptive factors (IL-1β, IL-33, IFNβ, NGF, BDNF, LIF and TSLP) and a decrease in a select set of anti-nociceptive factors (IL-12, IL-13, HGF and VEGF-A) could contribute to ocular surface discomfort without changes in tear fluid dynamics during the post face-mask-wearing period.

**Table 1 biomedicines-10-01160-t001:** Association between the levels of tear soluble factors and immune cell proportions on the ocular surface with ocular disease index profile.

	OSDI-Total		OSDI-Discomfort Scale		OSDI-Vision Scale	
	r	*p*-Value	r	*p*-Value	r	*p*-Value
Analytes						
IL-1β	0.310	0.010	0.279	0.021	0.191	0.119
IL-6	−0.279	0.021	−0.313	0.010	−0.175	0.154
IL-8	−0.297	0.014	−0.315	0.009	−0.157	0.202
IL-12p70	−0.378	0.002	−0.442	0.000	−0.157	0.200
IL-12/23p40	−0.198	0.105	−0.303	0.012	−0.072	0.561
IL-13	−0.279	0.021	−0.295	0.015	−0.194	0.113
IL-33	0.375	0.002	0.334	0.005	0.212	0.083
IFNα	−0.155	0.207	−0.231	0.058	−0.133	0.278
IFNβ	0.242	0.047	0.213	0.081	0.101	0.411
BDNF	0.384	0.001	0.359	0.003	0.216	0.077
Eotaxin	−0.158	0.199	−0.269	0.026	−0.043	0.730
Granzymes	−0.280	0.021	−0.281	0.021	−0.193	0.115
TSLP	0.307	0.011	0.278	0.022	0.181	0.140
VEGF	−0.275	0.023	−0.218	0.075	−0.171	0.164
sTNFRI	0.253	0.038	0.218	0.075	0.228	0.062
sFasL	−0.218	0.074	−0.285	0.019	−0.079	0.523
Immune cells						
Leukocytes	0.262	0.086	0.421	0.004	−0.059	0.704
Macrophage	0.376	0.012	0.327	0.030	0.366	0.015
NK cells	0.306	0.043	0.370	0.014	0.000	0.998
NKT cells	0.207	0.178	0.275	0.071	0.017	0.913

r = Spearman rank correlation coefficient; *p* < 0.05 is considered to be statistically significant.

**Table 2 biomedicines-10-01160-t002:** The levels of secreted factors in supernatant of corneoscleral rim explant exposed to hypercapnia.

Analytes (pg/mL)	Normocapnia (5% CO_2_)			Hypercapnia (20% CO_2_)			Fold Diff.
	Mean	Stdev	SEM	Mean	Stdev	SEM	
IL-1α	8.9	18.3	10.6	15.4	1.2	0.7	1.7
IL-6	4654.9	157.7	91.0	184.1	0.1	0.1	0.0
IL-8	328.8	28.9	16.7	163.8	0.4	0.2	0.5
IL-13	1.6	0.1	0.1	1.3	0.1	0.0	0.8
IFNβ	17.3	35.1	20.3	43.6	2.2	1.3	2.5
TNFα	2.1	0.1	0.0	0.5	2.4	1.4	0.2
HGF	9573.9	4395.3	2537.6	5961.2	1.0	0.6	0.6
VEGF	422.0	67.7	39.1	267.6	0.6	0.4	0.6
LIF	448.5	191.6	110.6	518.6	0.3	0.2	1.2
Perforins	5.9	2.4	1.4	8.4	0.8	0.4	1.4

*n* = 3; Wilcoxon matched-pairs signed rank test; no statistical significance was observed; concentration in pg/mL normalised to total protein concentration for each sample.

**Table 3 biomedicines-10-01160-t003:** (**a**) Status of mucin 5AC and mucin 16 in tear fluid in study subjects following mask wear and in corneoscleral explant culture following exposure to hypercapnia. (**b**) Status of mucin 5AC and mucin 16 in corneoscleral explant culture following exposure to hypercapnia.

(**a**)		
**Tear Fluid**	**MUC5AC**	**MUC16**
	**Fold Diff.** (**Post-ME/Pre-ME**)	**Fold Diff.** (**Post-ME/Pre-ME**)
Subject 1	1.6	1.3
Subject 2	4.9	0.8
Subject 3	0.5	7.5
Subject 4	0.5	0.7
Subject 5	0.1	0.9
Subject 6	1.8	1.4
Mean ± SEM	1.6 ± 0.7	2.1 ± 1.1
(**b**)		
**Corneoscleral Rim Explant Culture**	**MUC5AC**	**MUC16**
	**Fold Diff.** (**Hypercapnia/Normocapnia**)	**Fold Diff.** (**Hypercapnia/Normocapnia**)
Donor 1	5.2	0.8
Donor 2	4.4	3.4
Donor 3	1.8	2.1
Mean ± SEM	3.8 ± 1.0	2.1 ± 1.1

Wilcoxon matched-pairs signed rank test; no statistical significance was observed.

## Data Availability

Data will be available on request.

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
