# Peer review of "Altered Ocular Surface Health Status and Tear Film Immune Profile Due to Prolonged Daily Mask Wear in Health Care Workers"

_biomedicines, 2022, doi:10.3390/biomedicines10051160_

Round 1

Reviewer 1 Report

Dear authors:

During the attack of COVID-19,  We are luck to learn more knowledge which          benefit for all the readers.  This article is very good.

However, I have a question about the detailed method of collection of the              tear.; We could not know how to collect the tear fluid from line 131 to            136.  We know the difficulty in collect the tear including the technique            and the volume of tear fluid for further study in our research. Could                you mention it ? I believe that the manuscript would be excellent after            your working hard.

Many thanks.      

Author Response

We thank the reviewer for the encouraging words and this suggestion. We have included the following details “Tear fluid samples were collected from the study subjects using Schirmer's strips by following Schirmer's test I protocol. Briefly, the blunt end of the sterile Schirmer’s strip was folded at the notch by 90 degrees to form a hook-shaped bend. The bent end of a Schirmer’s strip was placed gently into the inferior-temporal aspect of the conjunctival sac (lower eye lid) of the subject’s eye using sterile forceps. The Schirmer’s strip was then al-lowed to wet by subject’s tear fluid via capillary action. The wetting length in millimeters were measured by observing the wetting end against graduation/scale marks on the strip at the end of 5 minutes. The Schirmer’s strip was the collected and stored in a sterile microcentrifuge tube at −80 °C until further processing.” in the relevant method sub-section (line no. 132 – 140) in the revised manuscript.

Reviewer 2 Report

This study evaluated possible effects of prolonged mask wearing on the ocular surface and tear proteins. In vitro studies using elevated CO2 levels was also performed. They found an overall increase in certain pro-inflammatory cytokines at the second timepoint in vivo. Overall, this is a well-written paper with interesting findings. Acute versus chronic effects should be clarified in the discussion since some of the tear findings may be a function of acute mask wearing, rather than chronic changes. Inclusion of a separate cohort to test the effect of acute mask wearing would be needed to address this question. Suggestions are made to improve the quality of the manuscript.

-Please provide the mean age and standard deviation (range) and sex distribution of study volunteers within the methods. 

-Please specify in the Methods if tears isolated at the 6 month timepoint were immediately following mask wearing. Some of the findings may be an acute response to wearing a mask, rather than prolonged changes to the ocular surface. Please clarify in the manuscript. The limitation of not including a non-mask wearing group (understandably) for comparative analysis should be noted in the discussion.

-IL-6 does not appear to be reduced in the Hyp. cap group (lines 359-361). Please clarify in the Results as well as the discussion since the findings are not statistically significant for many of these factors. The small sample size is a limitation.

-Table 2 legends states that no statistical differences were present. Please clarify in the text (lines 363-366).

Author Response

This study evaluated possible effects of prolonged mask wearing on the ocular surface and tear proteins. In vitro studies using elevated CO2 levels was also performed. They found an overall increase in certain pro-inflammatory cytokines at the second timepoint in vivo. Overall, this is a well-written paper with interesting findings. Acute versus chronic effects should be clarified in the discussion since some of the tear findings may be a function of acute mask wearing, rather than chronic changes. Inclusion of a separate cohort to test the effect of acute mask wearing would be needed to address this question. Suggestions are made to improve the quality of the manuscript.

Authors’ response: We thank the reviewer for his comments and insight. We have addressed all the comments pointwise below and made the necessary changes to the manuscript.

-Please provide the mean age and standard deviation (range) and sex distribution of study volunteers within the methods. 

Authors’ response: Thanks for the suggestion. We have included the details “Age 30±4.2 years; M/F – 9/8” in line no. 103 in the method section of the revised manuscript.

-Please specify in the Methods if tears isolated at the 6-months timepoint were immediately following mask wearing. Some of the findings may be an acute response to wearing a mask, rather than prolonged changes to the ocular surface. Please clarify in the manuscript. The limitation of not including a non-mask wearing group (understandably) for comparative analysis should be noted in the discussion.

Authors’ response: We understand the basis of reviewer’s questions. The sample collections both during Pre-face mask wearing period and post-face mask wearing period (6 months timepoint) were conducted between 10 am and 12 noon. It is important to note that the 6 months timepoint sample collection was conducted during the active COVID-19 pandemic when all were trained to keep face masks on continuously everyday. The sample collection was conducted immediately after their arrival at the hospital, within two hours of face mask wear for the day. The reviewer’s point about the effects observed being due to acute response of face mask wear may apply, although we surmise that such acute effects would be observed if the samples and clinical parameters were measured at the end of the day i.e., a minimum of an 8-hour period of face mask wear. Thus the clinical and molecular changes observed could possibly be predominantly due to the effect prolonged wear of face mask. We also appreciate the reviewer’s understanding of the unavailability of additional matched (pre- and post- six months) non-mask wearing group as control group due to the pandemic. Therefore, we have now included a sentence stating this as a potential limitation of the study (lines 528-531: “The limitations of the study are the lack of a control group without mask wear, which was not possible in view of the health hazard and multiple time points of measurement to delineate acute vs chronic responses.”). 

Since, the reviewer has raised an important aspect and in order to improve the clarity for the readership, the following amended sentence “Clinical parameters included ocular surface disease index (OSDI) score, Schirmer’s test-1 (ST1), tear break up time (TBUT), tear film interferometry using the Lipiview (TearScience Inc) and objective scatter index (OSI) on the Optical Quality Analysis System II (OQAS, Vi-siometrics, Terrassa, Spain) which were measured along with collection of tear fluid and ocular surface wash from the same subjects during Pre-FM and Post-FM between 10am and 12 noon.” in line 115 – 120 has been included in the method section of the revised manuscript.

-IL-6 does not appear to be reduced in the Hyp. cap group (lines 359-361). Please clarify in the Results as well as the discussion since the findings are not statistically significant for many of these factors. The small sample size is a limitation.

Authors’ response: We thank the reviewer for this suggestion. We have made the necessary amends as follows “The expression of inflammatory factors such as IL-8 and TNFα were reduced post hypercapnia (Figure 5d – f) while IFNβ was elevated (Figure 5g), although the expression of IL-6 and VEGF-A were not altered (Figure 5d and 5h). It is to be noted that despite the trend that was observed, only reduction in expression of IL-8 was statistically significant” in line no. 363 – 367 in the relevant section of the revised manuscript. In addition, as per reviewer suggestion, we have included the following sentence “The lack of statistically insignificance in these explant culture experiments could be possibly due to small number of corneoscleral rims used.” in line no. 506 – 508 in the discussion section of the revised manuscript.

-Table 2 legends states that no statistical differences were present. Please clarify in the text (lines 363-366).

Authors’ response: We agree with the reviewer’s recommendation. The sentence was amended as follows “We observed IL-6, IL-8, TNFα, HGF and VEGF levels to be reduced but IL1α, IFNγ, LIF and Perforins levels to be elevated, albeit statistically insignificant, in hypercapnic samples compared to normocapnic controls (Table 2), similar to that observed in the tear samples (Figure 2-3).” in line no. 369 – 372 in the relevant section of the revised manuscript.